# Perspectives on Berberine and the Regulation of Gut Microbiota: As an Anti-Inflammatory Agent

**DOI:** 10.3390/ph18020193

**Published:** 2025-01-31

**Authors:** Quintero Vargas Jael Teresa de Jesús, Juan-Carlos Gálvez-Ruíz, Adriana Alejandra Márquez Ibarra, Mario-Alberto Leyva-Peralta

**Affiliations:** 1Department of Health Sciences, University of Sonora, Cd. Obregón 85010, Mexico; adriana.marquez@unison.mx; 2Department of Chemical and Biological Sciences, University of Sonora, Hermosillo 83000, Mexico; juan.galvez@unison.mx; 3Department of Chemical-Biological and Agricultural Sciences, Universidad de Sonora, Unidad Regional Norte, Caborca 83621, Mexico; mario.leyva@unison.mx

**Keywords:** berberine, gut, dysbiosis

## Abstract

Berberine is a promising agent for modulating the intestinal microbiota, playing a crucial role in human health homeostasis. This natural compound promotes the growth of beneficial bacteria such as *Bacteroides*, *Bifidobacterium*, and *Lactobacillus* while reducing harmful bacteria such as *Escherichia coli*. Clinical and preclinical studies demonstrate that Berberine helps regulate T2D and metabolic disorders, improves blood glucose levels during T2D, and reduces lipid profile and chronic inflammation, especially when combined with probiotics. Berberine represents a promising adjuvant therapy for inflammatory diseases, particularly intestinal disorders, due to its multifaceted actions of inhibiting proinflammatory cytokines and pathways during IBS, IBD, and UC and its modulation of gut microbiota and/or enhancement of the integrity of the intestinal epithelial barrier. This review establishes the basis for future treatment protocols with berberine and fully elucidates its mechanisms.

## 1. Introduction

### 1.1. Berberine

Berberine, 9,10-dimethoxy-5,6-dihydro[1,2]dioxolo[4,5-g]isoquinolino[3,2-a]isoquinolin-7-ium (C_29_H_19_NO_4_), is an isoquinolinic quaternary alkaloid with a molar mass of 337.37 g/mol [1,2]. It is slightly soluble in cold water and ethanol and moderately soluble in methanol [3]. It has a heterocyclic structure formed by four rings of six members and a methylenedioxy group (A, B, C, and D). The five-member acetal, has been associated with various biological activities. The C ring has a quaternary ammonium (N+) strongly associated with antimicrobial and anticancerous activities [4]. The C9 and C10 carbons in the D ring bear a methoxy group, especially attractive for metabolic regulation activities such as lipid and glycemia [4]. Berberine is a characteristic fluorescent yellowish coloring agent used in the industry as a textile dye (wool, leather) [5,6]. It is classified as a natural yellow dye 18 (C.I. 75160) belonging to the group of pyridine-based dyes, the only natural dye in this class [7,8] (Figure 1).

Berberine is an alkaloid isolated from various plants in the Berberis genus, including Berberis aquifolium, Berberis vulgaris, and Berberis aristata [9]. It can also be extracted from other plants such as Coptis chinensis, Rhizoma coptidis [10], Arcangelisia flavar, Argemone mexicana, Argemone gracilenta, Hydrastis canadensis, Phellodendron amurense, and Phellodendron chinense [8,10,11]. Berberine is found in higher concentrations in these plants’ bark, leaves, branches, rhizomes, roots, and stems [3,6,8].

The most common extraction methods involve using various solvents, both polar and non-polar. Techniques such as maceration, sonication, soxhlet extraction, and water heating are employed, depending on the polar characteristics of the compounds. These methods produce a heterogeneous extract of chemical compounds, which can then be separated and analyzed through techniques like thin-layer chromatography (TLC), mass spectrometry (MS/MS-HPLC), and various spectroscopic methods such as nuclear magnetic resonance (NMR), infrared (IR), ultraviolet (UV), and Fourier transform infrared (FTIR) spectroscopy, among others, based on their intended applications [1].

The Asia–Pacific region, particularly China, is the leading area for producing and supplying berberine. While global production of berberine is fluctuating, it is on an upward trend [12] due to rising demand for pharmaceuticals, nutraceuticals, and functional foods. The global berberine market was valued at USD million in 2023 and is projected to grow by 2030. This growth is primarily driven by its increasing use in treating various conditions, including metabolic disorders, inflammation, and cancer [13].

Initially investigated for its antimicrobial and anti-inflammatory properties in treating infectious bacterial and gastrointestinal diseases [14,15], berberine has demonstrated a broad range of pharmacological activities, including antidiabetic [16,17] and antioxidative effects [17]. Recently, it has also been explored for anticancer activity in cellular and animal models. In vitro studies show that berberine can inhibit cell growth and induce cell cycle arrest and apoptosis in various cancer cell lines, including prostate, gastric, liver, cervical, and breast cancer cells, as well as skin and hematological cell lines [18]. Additionally, it helps protect the intestinal epithelial barrier from injury [19]. Berberine has recently been identified as a promising candidate for modulating the intestinal microbiota and maintaining inflammation homeostasis in chronic inflammatory diseases [20,21,22,23,24,25,26,27]. The diverse pharmacological properties of berberine suggest it has significant potential as a drug with a broad spectrum of clinical applications.

### 1.2. Traditional Use of Plant Species Containing Berberine

The primary natural source of berberine is plants from the Berberis genus. These plants have a long history of use for treating various health issues, including inflammation, infectious diseases, diabetes, constipation, and infections of the ear, eye, and mouth. They have also been used to treat hemorrhoids, uterine and vaginal conditions, toothaches, and asthma [28,29]. The earliest documentation of using the fruit of Berberis vulgaris as a blood-purifying agent dates back to clay tablets found in the library of the Assyrian Emperor Ashurbanipal, around 650 BC. In Asia, berberine-rich plants—mainly the stem, bark, roots, and root bark—have been a cornerstone of traditional Chinese and Ayurvedic medicine for over 3000 years [1,29]. Traditionally, extracts and decoctions from plants containing berberine have been employed against various microorganisms, including bacteria, viruses, fungi, and parasites [30].

Plants from the Mahonia genus that contain berberine have been identified, with Mahonia aquifolium being one of the most notable. This plant has been traditionally used to treat various skin infections and is recognized in traditional Asian medicine for its antimicrobial properties. In traditional Chinese medicine, *Coptidis rhizoma*, which contains berberine, is used to eliminate toxins [1].

Currently, dietary supplements derived from plants rich in berberine are employed to address respiratory issues such as the common cold, infections, and flu and help reduce fever [1,31,32]. Positive effects have been observed on the mucous membranes of the upper respiratory tract and the gastrointestinal system, showing benefits for conditions associated with these areas [1,33].

## 2. Gut Microbiota

The microbiota is essential to human health, as bacteria are present from birth through development. Contact with the vaginal canal during childbirth, breastfeeding, and teething are important factors that favor bacterial growth in different tissues, such as the respiratory tract, gastrointestinal tract, and genitourinary tract [34]. Due to a symbiotic relationship developed over millions of years, the intestinal microbiota intervenes in several processes that impact the physiology of human beings. Human intestinal colonization is considered one of the most relevant functions of intestinal microbiota. It promotes catalytic functions that directly affect the host metabolism, thus allowing the generation of simplified macromolecules more easily absorbed by enterocytes. In addition to the protection concerning the barrier effect that it induces by covering the extensive surface area of the small and large intestines, it prevents the colonization of pathogenic organisms and the development of infections. It is also observed that the microbiota presents trophic effects strongly related to the proliferation and differentiation of tissues, thus favoring the growth of immunological tissues associated with mucosae, particularly intestinal mucosae (MALT and GULT) [34,35]. The latter is made up of cellular components such as enterocytes, M cells, goblet cells, and intraepithelial lymphocytes (IEL, CD8^+^), among others, as well as proteins such as defensins, antimicrobial peptides and mucosal secretion antibodies such as IgA. The GULT and its elements primarily neutralize and eliminate antigens of systemic origin, food, and even those associated with the intestinal microbiota. It also maintains a homeostasis process, ensuring a symbiotic relationship between the intestinal microbiota and all its elements. A chronic or exacerbated intestinal immune reaction in these tissues can lead to the development of autoimmune diseases [34,35].

The intestinal microbiota comprises various microorganisms such as bacteria, fungi, and even entities like viruses. Bacteria are vital in humans’ homeostasis of many physiological, metabolic, and immunological processes. They are found in great diversity within the tissue of the small and large intestine [35]. Intestinal bacteria can be found from the proximal duodenum to the terminal ileum in an approximate range of 10^4^–10^7^ bacteria/mL and the colon 10^11^–10^12^ [36,37]. The most dominant species in the intestinal microbiota are *Bacterioidetes*, which mainly contain the genera *Bacteroides* and *Prevotella*; *Firmicutes*, consisting of more than 200 different genera, among which are *Lactobacillus*, *Bacillus*, *Enterococcus*, *Ruminicoccus*, and *Clostridium*; and *Actinobacteria*, *Proteobacteria* and *Fusobacteria* [38]. Among the strains most frequently isolated in feces in healthy human adults or through bacterial identification by 16S rRNA sequencing are *Bacteroides*, *Clostridium leptum*, *Eubacterium*, *Bifidobacterium*, *Enterobacteriaceae* such as *Escherichia coli*, *Proteus*, *Klebsiella*, and *Lactobacillus*, among others [36,37,38].

The varied composition of the bacterial microbiota in humans may be due to various factors such as gestational age at birth (newborn, preterm), type of birth, forms of breastfeeding, age, pharmacology, body mass index (BMI), cultural and dietary habits, ethnicity, exercise, immunological competence associated with the development of autoimmune diseases, and genetic susceptibility. Even the emotional state strongly influences the induction of dysbiosis [34,35,36]. Therefore, the presence of beneficial bacteria such as *Bifidobacteria* and *Bacteroides* spp., as well as the frequency of detection of *Firmicutes* and *Bacterioidetes* in inflammatory processes, obesity, and even psychiatric diseases, has led to their classification as “bad” strains. Alterations in the intestinal microbiota or dysbiosis can be caused by genetic and individual-specific factors, as well as behavioral factors related to eating habits, with evident changes in the intestinal microbiota being observed when there are changes in the incorporation of macronutrients into the diet. Additionally, intestinal bacterial disorders can lead to an imbalance in the regulation of the immune response, affecting the integrity of the epithelial barrier and causing inflammatory events, which can lead to metabolic diseases, chronic inflammatory diseases, and cancer, among others.

The pursuit of new drugs to combat intestinal dysbiosis is a global priority, and natural products, particularly alkaloids, have emerged as promising sources of novel chemical synthons with potent and effective biological effects. The potential of alkaloids in drug development is significant and warrants further exploration. In nature, around 40,000 alkaloids with diverse structural characteristics are known. Some members of this family have much pharmacological relevance [39,40]. Alkaloids can be categorized into three groups according to their chemical precursors during plant synthesis: true alkaloids, protoalkaloids (both derived from amino acid metabolism), and pseudoalkaloids (not derived from amino acids) [14,15,41,42]. They present a broad spectrum of biological activities such as antimicrobial [43,44,45], anti-viral [45,46], anti-diabetic [47], and anticancer [47,48,49,50]. Moreover, they have been widely associated with improving biochemical factors in nervous, cardiac, and chronic degenerative diseases, such as diabetes and hypertension [51,52,53]. They have also been linked to controlling diarrhea in patients with irritable bowel syndrome and improving symptoms of depression and anxiety [54].

### 2.1. Berberine and Gut Microbiota

Berberine induces changes in the intestinal microbiota by promoting the frequency of beneficial bacteria in the intestine, such as *Bacteroides*, *Bifidobacterium*, *Lactobacillus*, and *Akkermansia* [20,21]. These are generally determined by studying changes in fecal metabolites by 16S rRNA and corroborated by fecal microbiota transplantation (FMT) [20]. In rats with mucositis in response to 5-Fluoroacyl, berberine enriched the relative abundance of *Firmicutes* and decreased *Proteobacteria*. In a study involving mice treated with berberine, researchers observed an increase in the abundance of *Verrucomicrobiota*, as well as the genera *Porphyromonadaceae*, *Akkermansia*, *Parabacterioides*, and *Lachnospiraceae* (specifically Lactobacillus). There was also an increase in *Clostridiales*, *Ruminococcus*, *Prevotella*, and *Clostridium IV*, along with decreased levels of *Escherichia*, *Alistipes*, and *Clostridium*. These changes may be attributed to the effects of berberine. Furthermore, mice treated with berberine or traditional Chinese medicine formulas like Gegen Qinlian Decoction showed an increase in the frequency of *Faecalibacterium*, *Roseburia*, *Clostridium XIVa*, *Ruminococcus2*, and *Dorea*, which are bacteria known for butyrate production, as well as an enrichment of less frequent groups such as *Parabacteroides*, *Paraprevotella*, *Butyricimonas*, *Alistipes*, *Gemmiger*, *Butyricicoccus*, and *Coprococcus* [22].

### 2.2. Berberine as a Prebiotic Agent

Even though there are many studies on the biological activities of bacteria, eukaryotic cells, and parasites, there are only five clinical trials in humans developed on the use of berberine as a prebiotic agent and/or in combination with probiotics and its impact on the intestinal microbiota in various pathologies. Ming et al. conducted a study between October 2015 and April 2018 with the objective to evaluate the effectiveness and safety of the combination of Bifidobacterium and berberine in reducing hyperglycemia and its regulatory impact on the intestinal microbiota. In this study, 300 participants aged 18–70 years diagnosed with hyperglycemia were randomly assigned to one of four groups: berberine (Be), *Bifidobacterium* (Bi), a combination of both (BB), and a placebo. Over 16 weeks, the subjects in the berberine groups (Be and BB) experienced a significant reduction in fasting plasma glucose levels, while those in the placebo group showed an increase. The BB group also saw a substantial decrease in 2-h postprandial glucose (2-h PPG) and glycated hemoglobin (HbA1c). The study also found that berberine led to changes in the composition of intestinal bacteria, including an increase in the abundance of *Ruminococcus gnavus* and *R. torques* in the Be group, which correlated with improvements in glucose regulation and insulin sensitivity. Additionally, the BB group had a lower increase in harmful bacteria such as *Klebsiella pneumoniae*, indicating that Bifidobacterium may help counteract potential side effects of berberine (see Table 1) [23].

Wang et al. conducted a study between August 2016 and July 2017 in 20 medical centers in China, aiming to evaluate the effectiveness of a combination of berberine and probiotics in improving postprandial hyperlipidemia. This study involved 365 newly diagnosed patients with type 2 diabetes, aged between 20 and 70 years and with a BMI of 19.0 to 35.0 kg/m^2^. Patients were divided into four groups: one group received a combination of berberine and probiotics (Prob + BBR), another group received berberine alone, the third group received probiotics alone, and the fourth group received a placebo. The intervention lasted 13 weeks and involved administering 0.6 g of BBR twice daily and multispecies probiotics, including *Bifidobacterium* breve and *B. longum*, once daily. Microbiological analyses showed significant differences in gut microbiome composition between the groups. The Prob + BBR group showed considerable restoration of *B. breve*, which improved postprandial total cholesterol and low-density lipoprotein cholesterol levels. Conversely, the group receiving only BBR showed reduced *Bifidobacterium* diversity, limiting its effectiveness in lowering postprandial lipids. Additionally, the Prob + BBR group showed an increase in the abundance of *Eggerthella lento*, which was positively correlated with postprandial lipid levels, suggesting its involvement in the modulation of postprandial lipidemia (Table 1) [24].

In a separate study, Zhang et al. (2016–2017) examined the impact of berberine and probiotics on the gut microbiota of individuals with type 2 diabetes. The study involved 209 newly diagnosed patients with type 2 diabetes, aged 20 to 70 years, and with a BMI between 19.0 and 35.0 kg/m^2^. The participants were randomly divided into four groups: berberine (BBR), probiotics with berberine (Prob + BBR), probiotics alone (Prob), and placebo (Plac). The intervention lasted 12 weeks and included administering BBR at a dose of 0.6 g twice daily before meals and probiotics of 4 g once daily in the evening. The findings revealed that the groups receiving BBR experienced a significant reduction in HbA1c levels compared to the placebo and probiotics alone groups. Additionally, metagenomic analyses demonstrated notable changes in the intestinal microbiota, including a decrease in species such as *Ruminococcus bromine* and alterations in bile acid metabolism (Table 1) [25].

Pu et al. conducted a study between May 2018 and April 2020 in five hospitals in China, which aimed to investigate the effects of berberine on the gut microbiota in patients with olanzapine-induced mild metabolic disorders. This study included 132 patients aged 16–65 years with schizophrenia, bipolar disorder, or schizoaffective psychosis, all treated with olanzapine for at least nine months. Patients were randomly assigned to receive either berberine or a placebo for 12 weeks, and the intervention consisted of berberine administered at doses of 0.1 g to 0.3 g three times daily, while the control group received a placebo. After 12 weeks, patients treated with berberine presented a significant decrease in fasting plasma glucose (FPG), HbA1c, triglycerides (TG), BMI, and waist circumference compared to the placebo group (*p* < 0.05). Regarding the intestinal microbiota, significant differences were found between the groups: the abundance of *Firmicutes* in the berberine group after treatment (8.85 × 10^7^ ± 5.36 × 10^7^ cp/μL) compared to the placebo group (3.47 × 10^8^ ± 1.84 × 10^8^ cp/μL, *p* < 0.01). Furthermore, coliform abundance was lower in the berberine group (2.58 × 10^6^ ± 8.33 × 10^6^ cp/μL) compared to the placebo group (3.11 × 10^6^ ± 1.76 × 10^6^ cp/μL, *p* < 0.05). In contrast, *Bacteroides* abundance was significantly increased in the berberine group (2.19 × 10^7^ ± 1.27 × 10^7^ cp/μL) compared to the placebo group (1.68 × 10^7^ ± 6.88 × 10^6^ cp/μL, *p* < 0.05). These results suggest that berberine modulates the gut microbiota, which might be related to its metabolic benefits in patients with olanzapine-induced metabolic disorders (Table 1) [26]. 

Finally, Li et al. conducted a study in China between April and June 2021 aiming to investigate the effects of berberine hydrochloride on gut microbiota and inflammatory cytokine levels in 68 patients with Parkinson’s disease, aged between 55 and 75 years. The patients were randomly divided into two groups: a control group that received conventional treatment and an observation group that, in addition to traditional therapy, received berberine hydrochloride at a dose of 0.2 g three times a day for three months. The results showed a significant reduction in inflammatory cytokine levels in the observation group compared with the control group. For example, IL-8 levels decreased from 62.26 ± 10.26 µg/L to 42.35 ± 10.24 µg/L in the observation group, whereas in the control group, they decreased from 60.29 ± 11.57 µg/L to 47.88 ± 10.25 µg/L (*p* < 0.05). Regarding intestinal microbiota diversity, the Chao and Ace indices increased significantly in the observation group after treatment (*p* < 0.001). In contrast, the changes in the control group were less pronounced (*p* < 0.05) [27]. The summary of the findings on intestinal microbiota is shown in Table 1.

The reviewed studies support using berberine as a prebiotic agent for regulating intestinal microbiota, especially when combined with probiotics in metabolic and inflammatory conditions. The results consistently demonstrate that berberine can alter the bacterial composition of the intestine, promoting the growth of beneficial microorganisms such as *Ruminococcus gnavus* and *Bifidobacterium* breve while reducing the presence of pathogens such as *K. pneumoniae*. Moreover, these changes in the microbiota are associated with significant improvements in metabolic parameters such as fasting plasma glucose, glycated hemoglobin, and postprandial lipid levels. However, the variability in doses and combinations used and in the populations studied suggests the need for further research to establish standardized protocols that optimize the benefits of berberine in different clinical contexts.

## 3. The Potential of Berberine in Treating Gut Inflammatory Disease Is a Significant Area of Research That Holds Promise for the Future of Gut Health

The small intestine is a crucial organ responsible for digesting and absorbing nutrients, and it also plays a significant role in disease development. Disruption of its functions can lead to diseases, with inflammatory intestinal diseases being the most common. These diseases share symptoms such as inflammation in the gastrointestinal tract and can be influenced by factors including family history, environment, diet, smoking, medication, and changes in the intestinal microbiota. Modification of the intestinal bacterial composition is a critical factor in the development of diseases such as irritable bowel syndrome (IBS), inflammatory bowel disease (IBD), ulcerative colitis (UC), Crohn’s disease, and type 1 diabetes mellitus (T1D). Some allergic and autism conditions are linked to intestinal dysbiosis. Current treatments focus on symptom reduction rather than addressing the underlying issue. Therefore, the study of new drugs from natural products, such as berberine, and their impact on modulating the inflammatory response and bacterial composition is becoming increasingly important.

Berberine is an alkaloid that has been used in the treatment of intestinal diseases, such as diarrhea [19,55,56]. Berberine induces the regulation of the intestinal microbiota; many studies reveal that BBR and some derivatives improve dysbiosis in humans with ulcerative diseases, as well as in rats with ulcerative diseases [21], modulating the activation of microglial cells [56], and decreasing intestinal inflammation generated in some cases by the disruption of the microbiota [55,57,58]. Additionally, dietary supplementation of berberine improves the gastrointestinal microbiota by enriching the members of the *Enterobacteriaceae* family and reducing the families *Ruminococcaceae*, *Lachnospiraceae*, and *Peptostreptococcaceae*, improving intestinal crypts, as well as T lymphocyte infiltration [19,58]. Therefore, it is of great interest to study the physiological–immunological relationship of berberine with intestinal dysbiosis and autoimmune diseases associated with intestinal tissues is of great interest.

### 3.1. Irritable Bowel Syndrome

Irritable bowel syndrome (IBS) is the most studied chronic gastrointestinal disorder. It manifests a great diversity of symptoms that can vary in intensity and frequency between patients. The most common symptoms are abdominal pain and cramps, bloating and gas, diarrhea, constipation, and the presence of mucus in stool. It can accompany fatigue, nausea, general pain, and sleep problems. It is characterized by altered intestinal motility, inflammation, visceral hypersensitivity, increased intestinal permeability, and alterations in the intestinal microbiota. Additionally, individuals with personality disorders such as depression have presented similar alterations in the fecal microbiota as in diarrheal patients with IBS-D [59].

Colonoscopy studies on adults diagnosed with IBS and treated with berberine show a decreased frequency of species belonging to the Firmicutes phyla. This suggests that these changes in the microbiota may affect the integrity of enterocytes in their cellular junctions, potentially leading to an inflammatory response [19,58,59,60,61,62]. Jia Q et al., 2019 found that the leading families of bacteria isolated in patients with IBS are *Bacteroides*-dominant type I or *Prevotella*-dominant type II, associated with chronic intestinal mucosa inflammation [21]. In an experimental model using rats (Sprague Dawley) transplanted with fecal microbiota (FMT) from healthy and IBS-positive patients, as well as those administered with berberine (200 mg/kg), hyperplasia in the liver of Kupffer cells (liver macrophages) and hepatic sinusoid hypertrophy were observed, along with elevated levels of proinflammatory hepatic cytokines such as TNF-α and IFN-β. The authors suggest that these conditions promoted the abundance frequency of *Faecalibacterium* and decreased *Bifidobacterium*, leading to alterations in the feces of these groups in formate, acetate, and propionate. It was also observed that berberine could have a reversible effect on macrophage hyperplasia. These findings contribute to understanding the pathogenesis of IBS-D and its relationship with the intestinal microbiota [21].

Li L. et al., 2020 developed berberine nanoparticles with baicalin to investigate the synergism of these molecules in mice with chronic stress-induced IBS-D plus senna decoction (0.6 g/kg). These nanoparticles showed a better therapeutic effect on visceral hypersensitivity and diarrhea in IBS-D model mice, compared with the BBR, BA, and BA/BBR mixture. In addition, BA-BBR nanoparticles reduced 5-hydroxytryptamine (5-HT), vasoactive intestinal polypeptide (VIP), and choline acetyltransferase (CHAT) in the colon. It reduced the expression of the transcription factor NF-kB, which promotes the synthesis of proinflammatory cytokines that target cell recruitment and increase inflammation. Therefore, this inflammation homeostasis at the intestinal microbiota level altered the presence of *Bacteroides*, *Deferribacteres*, *Verrucomicrobia*, *Candidatus*, *Saccharibacteria*, and *Cyanobacteria* in the intestinal microbiota [63].

### 3.2. Inflammatory Bowel Disease

Inflammatory bowel disease (IBD) is a chronic disease characterized by recurrent gastrointestinal complaints. It can be classified into ulcerative colitis (UC) and Crohn’s disease (CD), characterized by the development of inflammatory processes in various areas of the GIT. In particular, ulcerative colitis (UC) is a chronic, recurrent disease characterized by inducing ulcers in the mucosal wall of the rectum and colon. It can be derived from a dysregulated immune response, genetic susceptibility, and alterations in the intestinal microbiota, such as the loss of bacterial populations like *Lachnospiraceae* and *Bacteroides* [64,65]. In DSS-induced UC manipulation models, it has been observed that berberine improves dysbiosis in the colon, the tight junctions of enterocytes, and regulates the secretion of proinflammatory cytokines by inhibiting the PLA2-COX-2-PGE2-EP2 pathway, thus enhancing inflammatory processes in the colon [66]. After the elimination of the intestinal microbiota of mice with DSS-induced UC, the therapeutic effect of BBR disappeared, while fecal transplantation of mice from the BBR group significantly improved the inflammatory process characteristic of UC by regulating the expression of inflammatory cytokines such as TNF-α, IL-17A, and IL-22, and even modifies the proliferation of intraepithelial lymphocytes (ILC1, 2, 3). Berberine nanoparticles, with an average hydrodynamic size of 220 nm (230.2 ± 18.1 nm) and a relatively similar particle size distribution of 0.22, decreased the inflammatory response in DSS-induced ulcerative colitis by inhibiting the NFkB/STAT-3 pathway. They regulated the homeostasis of “harmful” microbiota such as *Enterobacteriaceae* and *Escherichia-shigella* and “beneficial” bacteria such as *Ruminococcaceae* and *Akkermansiaceae* [19,21,55,64,65,66,67,68,69,70,71].

Additionally, in DSS-induced UC in cats, berberine was found to regulate the intestinal microbiota, with an increase in beneficial microbiotas such as *Lactobacillacrae* and *Prevotellaceae* and an apparent reduction in *Bacteroidaceae* compared to the control group, as well as a decrease in symptoms such as inflammation and diarrhea. During UC induction by DSS in the cat model, an increase in *Bacteroides fragilis* and a reduction in *Lactobacillus ruminis* were observed, and after berberine, an increase in *L. ruminis* was found through 16S rRNA sequencing. Molecularly, berberine induces a notable decrease in inflammatory damage in tissues and even a reduction in the expression of proinflammatory cytokines such as IL-1β, IL-6, IL-8, IL-10, and TNF-α [21,72].

### 3.3. Berberine and Inflammatory Biomarkers

To determine whether berberine significantly affects proinflammatory biomarkers, a search was conducted for meta-analyses that included inflammatory biomarkers as primary or secondary outcomes. Five studies were identified that offer a comprehensive view of the effects of berberine on various inflammatory biomarkers and metabolic profiles. The survey by Beba et al. (2019) focused on evaluating the impact of berberine on C-reactive protein (CRP) levels and found a significant reduction of −0.64 mg/L (95% CI: −0.67 to −0.61) in patients with conditions such as acute coronary syndrome, hypertension, and type 2 diabetes. This substantial reduction in CRP levels, regardless of the dose used, provides reassurance about the effectiveness of berberine [73].

The study by Guo et al. (2021) provides strong support for the potential of berberine in managing type 2 diabetes and its inflammatory complications. Their research demonstrated significant reductions in various inflammatory and metabolic markers, including a decrease in glycated hemoglobin (HbA1c) by −0.75%, fasting glucose by −0.89 mg/dL, CRP by −2.13 mg/L, interleukin-6 (IL-6) by −1.83 pg/mL, and tumor necrosis factor (TNF-α) by −1.44 pg/mL. The study suggested that berberine is as effective as metformin with a favorable safety profile, highlighting its potential in managing type 2 diabetes and its inflammatory complications [74].

The study by Lu et al. (2022) further reinforces the potential of berberine in managing metabolic syndrome, particularly in specific populations such as the Chinese. Their meta-analysis on patients with metabolic syndrome observed that berberine significantly reduced CRP by −1.54 mg/L, TNF-α by −1.02 pg/mL, and IL-6 by −1.17 pg/mL levels. While no significant impact was observed on IL-1β levels, these findings suggest that berberine might be particularly useful in managing metabolic syndrome, especially in specific populations such as the Chinese [75].

In the context of acute ischemic stroke, Luo et al. (2023) evaluated the role of berberine as an adjuvant treatment, showing that it significantly reduces the levels of hs-CRP by −1.33 mg/L, Macrophage Migration Inhibitory Factor (MIF) by −2.00 ng/mL, and IL-6 by −2.94 pg/mL, among others detailed in Table 1. The authors highlighted that combining berberine with conventional treatment could improve the effectiveness rate due to its ability to reduce the levels of proinflammatory cytokines [76].

Finally, the study by Vahedi-Mazdabadi et al. (2023) focused on investigating the effect of berberine and barberry supplementation on inflammatory biomarkers and analyzing the dose–response relationship in adults. The research found that supplementation significantly reduced the levels of IL-6 by −1.18 pg/mL (95% CI: −1.8 to −0.55, *p* < 0.0001), TNF-α by −3.72 pg/mL (95% CI: −5.55 to −1.9, *p* < 0.0001), and CRP by −1.33 mg/L (95% CI: −1.9 to −0.75, *p* < 0.0001). Furthermore, a non-linear dose–response relationship was identified for IL-6 and TNF-α, with a significant decrease in serum levels with doses less than 1000 mg/day and an intervention duration of fewer than five weeks. Regarding serum CRP, a gradually decreasing trend was observed from 0 to 2000 mg/day, followed by an increase from 2000 to 10,000 mg/day. It was concluded that with doses up to 2000 mg/day, barberry and berberine supplementation had no notable effect on decreasing serum CRP concentrations [77]. Collectively, these studies underscore the potential of berberine in effectively reducing inflammatory biomarkers. With its favorable safety profile, berberine holds promise in managing chronic inflammatory and metabolic diseases. Refer to Table 2 for a summary of the studies.

### 3.4. Extraintestinal Disorders

Berberine’s pharmacological properties are wide-ranging. It has been studied for its potential in managing various intestinal and digestive inflammatory diseases and extraintestinal disorders like cardiovascular and neurological conditions. While historically used for gastrointestinal issues, recent research has also explored its effectiveness in treating extraintestinal diseases.

Berberine has shown promise in protecting intestinal epithelia, improving liver lesions, and mitigating acute ischemic renal injury due to reperfusion. It has also been linked to the treatment of allogeneic hematopoietic cell transplantation, where it was found to reduce inflammation and damage to target organs while modulating the intestinal microbiota. Additionally, berberine has been investigated for its potential in treating metabolic-related disorders such as T2D, dyslipidemia, and obesity, showing positive effects on glycemia regulation, lipid levels, and energy metabolism. Furthermore, research has indicated that berberine may improve musculoskeletal disorders, including osteoarthritis, and may enhance bone formation in postmenopausal women.

In cardiovascular diseases, berberine has been cardioprotective by improving cardiac function by improving cardiovascular hemodynamics and decreasing the development of atherosclerosis [78]; it improves blood circulation by regulating blood pressure [79,80]. In anti-inflammatory diseases such as RA, it inhibits the production of proinflammatory cytokines like IL-6 and TNF-α.

Berberine reduces neuroinflammation during intracerebral hemorrhage, improving neurological dysfunction by preventing the activation of proinflammatory pathways of CNS-resident macrophages, microglia [81], and encephalopathy [82].

In neurodegenerative diseases, the use of berberine has been related to the prevention and treatment of Alzheimer’s [83,84,85,86,87], Parkinson’s [80,84,87], and ischemic stroke due to its antioxidant and anti-inflammatory capacity. It protects neurons from oxidative damage and chronic stress. It modulates the intestinal microbiota to increase the abundance of butyric acid, which is related to improving cardiovascular events [88] and even psychiatric conditions such as anxiety [87,89,90,91]. It also decreases the effects of stress related to methamphetamine use through the reduction of locomotor activity compared to the methamphetamine control group [92].

The antiproliferative effect of berberine has been widely studied in various human and mouse cell lines, and even structural modifications have been reported through Structure-Activity Relationship studies, finding a significant improvement in activity [93,94,95]. Through the use of human melanoma cell lines [96], lung cancer [97], and triple-negative breast cancer [98], the antimetastatic or anti-invasive action of berberine has been analyzed through the inhibition of signaling pathways such as MAPK or the deregulation or inhibition of TGF-β production.

Thanks to its extensive pharmacological versatility, berberine has been identified as a promising adjunctive treatment for various extraintestinal diseases. Its ability to modulate multiple biological processes, including inflammation, oxidative stress, and energy metabolism, makes it an attractive option for chronic diseases such as diabetes, cardiovascular and neurodegenerative diseases, and cancer. However, the crucial need for further research to confirm its long-term safety and efficacy in humans cannot be overstated, underscoring the importance of ongoing investigation in the field.

## 4. Effects of Berberine on Obesity

Berberine has emerged as a promising therapeutic agent for obesity, with the potential to significantly impact public health. For instance, Xiong et al. (2020) conducted a dose–response meta-analysis that included nine randomized controlled trials (RCTs) to evaluate the effects of berberine supplementation on body mass index (BMI), waist circumference (WC), and body weight (BW) in adults. The study analyzed 983 individuals, 484 receiving berberine and 499 in the control group. The results showed a significant reduction in BMI, with a weighted mean difference (WMD) of −0.29 kg/m^2^ (95% CI: −0.51 to −0.08, *p* = 0.006). No significant heterogeneity was observed between the studies (I^2^ = 0.0%, *p* = 0.85). Additionally, berberine significantly reduced WC, with a WMD of −2.75 cm (95% CI: −4.88 to −0.62, *p* = 0.01), although notable heterogeneity was present (I^2^ = 90.6%, *p* = 0.001). However, no significant changes in BW were observed following supplementation (WMD: −0.11 kg, 95% CI: −0.99 to 0.76, *p* = 0.79) [99]. Subgroup analysis indicated that berberine was more effective in reducing WC in women (WMD: −2.45 cm, 95% CI: −3.48 to −1.43, *p* < 0.001), in individuals with an initial BMI > 30 kg/m^2^, with doses exceeding 1 g/day, and intervention durations longer than 12 weeks [99].

Ilyas et al. (2020) systematically investigated the effects of berberine on weight loss and obesity prevention. In preclinical models, berberine influenced gut microbiota by reducing microbial diversity at 100 mg/kg/day. It also inhibits α-glucosidase at 200 mg/kg/day and decreases adipocyte differentiation by lowering the expression of LXRs, PPARs, and SREBPs at 150 mg/kg/day. These effects contribute to reduced lipid accumulation and improved insulin sensitivity [100]. In human studies, berberine showed beneficial effects on the genetic regulation of cholesterol absorption and glucose accumulation at a dosage of 1 g/day. These findings indicate that BBR may effectively help prevent obesity when combined with lifestyle changes.

Asbaghi et al. (2020) conducted a meta-analysis including 12 studies to evaluate the effects of berberine on obesity parameters, inflammation, and liver enzymes. Their findings demonstrated that berberine supplementation led to significant reductions in body weight (WMD: −2.07 kg, 95% CI: −3.09 to −1.05, *p* < 0.001), BMI (WMD: −0.47 kg/m^2^, 95% CI: −0.70 to −0.23, *p* < 0.001), and WC (WMD: −1.08 cm, 95% CI: −1.97 to −0.19, *p* = 0.018). Additionally, berberine was found to reduce C-reactive protein (CRP), a marker of inflammation (WMD: −0.42 mg/L, 95% CI: −0.82 to −0.03, *p* = 0.034). However, there were no significant changes observed in liver enzymes, such as alanine aminotransferase (ALT) (WMD: −1.66 IU/L, 95% CI: −3.98 to 0.65, *p* = 0.160) and aspartate aminotransferase (AST) (WMD: −0.87 IU/L, 95% CI: −2.56 to 0.82, *p* = 0.311). This suggests that berberine does not negatively impact liver function [101].

Noh et al. (2022) investigated the molecular mechanisms behind the anti-obesity effects of berberine. By combining in silico and in vivo approaches, they identified that berberine influences the activity of adipose tissue macrophages (ATMs). This modulation reduces the infiltration of M1 proinflammatory macrophages and promotes the polarization of M2 anti-inflammatory macrophages. Their in silico analysis identified key therapeutic targets, described potential pathways, and simulated BBR docking at M1 and M2 macrophages and TNF-α, CCL2, CCL4, CCL5, and CXCR4. In their vivo experiment, 20 C58BL/6 mice were divided into four groups: standard chow, a high-fat diet (HFD) group, HFD + BBR (100 mg/kg) group, and HFD + metformin (200 mg/kg) group. They measured body weight, fat tissue area, glucose and fat tolerance, HOMA-IR, and changes in ATM populations [102]. The results showed that BBR significantly reduced body weight, adipocyte size, liver fat deposition, HOMA-IR, triglycerides, and free fatty acids. Additionally, it enhanced the population of CD206^+^ M2 ATM cells, which possess anti-inflammatory properties. These findings suggest that BBR alleviates obesity-induced inflammation by modulating ATM recruitment and polarization by inhibiting chemokine signaling [102].

These findings highlight BBR’s anti-obesity effects by inhibiting chronic inflammation, regulating glucose and lipid metabolism, and modulating macrophage recruitment through chemotaxis inhibition.

### 4.1. Effect of Berberine on Diabetes

In a comprehensive review, Askari et al. (2024) explored the effects of berberine on diabetes and its complications, highlighting its therapeutic potential [103]. The review indicates that berberine stimulates insulin secretion and enhances insulin resistance through multiple pathways. Specifically, it has been documented that berberine promotes the upregulation of protein expression of peroxisome proliferator-activated receptor (PPAR)-γ and glucose transporter (GLUT) 4. It activates the signaling pathways of PI3K/AKT and AMP-activated protein kinase (AMPK), contributing to glucose homeostasis and improving lipid metabolism. Berberine also exhibits significant protective effects against complications associated with diabetes, such as hepatic damage, cardiovascular disorders, nephropathy, and neuropathy. In animal models, berberine has been shown to reduce fasting glucose levels, improve lipid profiles, and decrease cardiac and renal fibrosis. For instance, in STZ-induced diabetic rats, berberine administration reduced inflammatory markers and increased AMPK activity, suggesting a mechanism by which berberine could alleviate chronic inflammation related to diabetes [103]. Additionally, berberine positively impacts gut microbiota, enhancing its anti-inflammatory and metabolic effects, indicating that its benefits extend beyond glucose regulation.

In a review conducted by Xie et al. (2022), a meta-analysis was carried out to investigate the effects of berberine on patients with type 2 diabetes. This analysis included randomized controlled trials where berberine was used as an intervention for individuals with type 2 diabetes, published up until November 2021. The study evaluated the hypoglycemic effects of berberine, focusing on key metrics such as fasting plasma glucose (FPG), glycated hemoglobin (HbA1c), and two-hour postprandial plasma glucose (2hPBG). The researchers used weighted mean differences (WMD) along with their 95% confidence intervals (CI) to analyze these effects [104].

To evaluate the safety of berberine treatment, we examined the incidence of adverse events and hypoglycemia using relative risk (RR) and its corresponding 95% confidence interval (CI). The meta-analysis included 37 studies with 3048 patients. The findings showed that berberine significantly reduced fasting plasma glucose (FPG) with a weighted mean difference (WMD) of −0.82 mmol/L (95% CI: −0.95, −0.70). It also reduced HbA1c, which had a WMD of −0.63% (95% CI: −0.72, −0.53), and 2-h postprandial blood glucose (2hPBG) with a WMD of −1.16 mmol/L (95% CI: −1.36, −0.96), all achieving statistical significance. Subgroup analyses indicated that the hypoglycemic effect of berberine was associated with the baseline levels of FPG and HbA1c in patients with type 2 diabetes. Furthermore, treatment with berberine—whether used alone or in combination with oral hypoglycemic agents—did not significantly increase the incidence of adverse events (RR = 0.73, 95% CI: 0.55, 0.97, *p* = 0.03) or the risk of hypoglycemia (RR = 0.48, 95% CI: 0.21, 1.08, *p* = 0.08) [104].

Berberine has demonstrated anti-diabetic activity in both in vivo animal models and human clinical trials. In liver cells, berberine lowers glucose levels by protecting against endoplasmic reticulum stress. It also stimulates insulin secretion from beta cells through its agonistic action on the GPR40 fatty acid receptor. Research indicates that berberine increases AMPK activity in L6 myotubes and 3T3-L1 adipocytes, which helps reduce lipid accumulation in these fat cells and enhances GLUT4 translocation in L6 cells. These effects are beneficial for managing diabetes and obesity. Additionally, berberine enhances the expression of insulin receptors in the liver by positively regulating insulin receptor mRNA and sustaining post-receptor signal transduction [1]. A study by Liu et al. revealed that berberine effectively addresses fat-induced hepatic insulin resistance in a type 2 diabetes hamster model by regulating various transcriptional pathways, including liver X receptor α (LXRα), sterol regulatory element-binding proteins (SREBPs), and peroxisome proliferator-activated receptor α (PPARα). However, due to its low bioavailability and poor absorption, berberine is thought to exert its anti-hyperglycemic effects primarily in the intestinal tract before reaching other organs, such as the pancreas and liver [1].

### 4.2. Toxicity of Berberine

Berberine exhibits varying levels of toxicity depending on the route of administration and dosage. High doses of oral berberine (100 mg/kg) have been linked to vomiting and can lead to the death of experimental animals after 8 to 10 days. In studies involving cats, oral administration of berberine sulfate also resulted in hemorrhagic symptoms. Mild adverse effects associated with berberine include salivation, diarrhea, nausea, vomiting, paralysis, and muscle tremors [105]. At subacute levels, it can cause gastric ulcers and liver and kidney problems, as indicated by elevated levels of liver enzymes (ALT and AST). Prolonged intraperitoneal administration of berberine has also been associated with teratogenic effects, atherosclerosis, and uterine contractions in animal models [1,105].

Regarding its immunotoxic and genotoxic effects, berberine has been shown to suppress both humoral and cellular immune functions. Studies with doses of 10 mg/kg revealed a reduction in leukocytes, lymphocytes, neutrophils, and spleen weight, along with a decrease in B and T cells—specifically, splenic CD4^+^ and CD8^+^ T cells and CD19^+^ B cells. On a genetic level, berberine can bind to nucleic acids, inhibiting cell differentiation and causing DNA damage. This explains its potential anticancer, neurotoxic, and phototoxic effects when administered at high doses or over prolonged periods [1,105]. These findings highlight the importance of rigorous safety measures in the clinical administration of berberine, particularly with intravenous or intraperitoneal routes.

## 5. Conclusions

The intestinal microbiota plays a crucial role in human health by regulating various physiological, metabolic, and immunological processes. Its influence begins at birth and is shaped by factors such as the type of delivery, breastfeeding, and diet. Berberine is known to modulate the intestinal microbiota by increasing the abundance of beneficial bacteria, which in turn improves metabolic health and reduces inflammation.

Furthermore, berberine has a positive impact on gut microbiota, reinforcing its anti-inflammatory and metabolic effects. This suggests that its benefits extend beyond just glucose regulation during antidiabetic treatments. When combined with probiotics, berberine appears to enhance its positive effects, indicating that it could function as a prebiotic agent in various clinical conditions. For instance, in cases of irritable bowel syndrome (IBS) and inflammatory bowel disease (IBD), including ulcerative colitis (UC), it helps reduce inflammation and improve the composition of intestinal microbiota, leading to decreased symptoms.

Berberine is also effective in treating metabolic diseases such as type 2 diabetes (T2D), dyslipidemia, cardiovascular diseases, and neurodegenerative disorders like Alzheimer’s and Parkinson’s disease. It lowers inflammatory biomarkers, including C-reactive protein (CRP), interleukin-6 (IL-6), and tumor necrosis factor-alpha (TNF-α), highlighting its anti-inflammatory potential in metabolic and inflammatory conditions.

Regarding safety, the toxicity of berberine varies with the method of administration and dosage. High oral doses (100 mg/kg) can induce vomiting and may be fatal in experimental animals. Mild side effects such as salivation, diarrhea, nausea, vomiting, and paralysis can occur. Prolonged intraperitoneal administration has been associated with teratogenic effects, atherosclerosis, and uterine contractions. Additionally, berberine can suppress immune functions and cause DNA damage.

Further clinical studies are necessary to standardize the use of berberine across different populations and diseases. While berberine shows significant potential in modulating intestinal microbiota and improving metabolic parameters—especially when combined with probiotics—additional research is needed to optimize its clinical applications.

## Figures and Tables

**Figure 1 pharmaceuticals-18-00193-f001:**
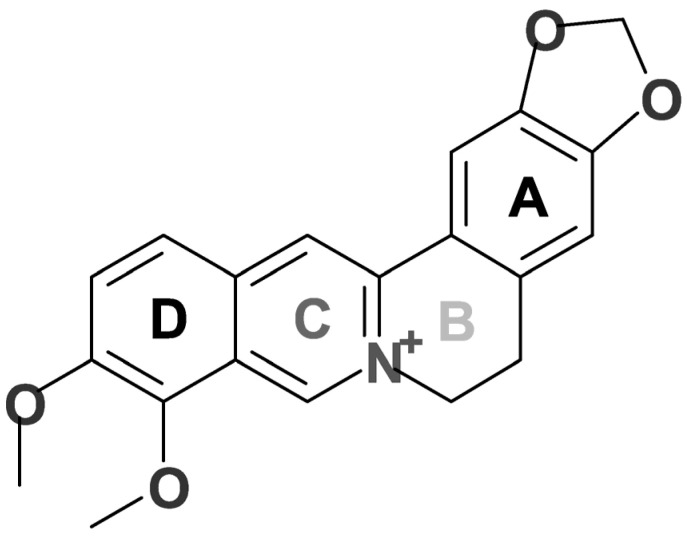
Chemical structure of berberine, by ChemDraw 23.1.1.3. The counterion is omitted.

**Table 1 pharmaceuticals-18-00193-t001:** Clinical studies on the impact of berberine on gut microbiota.

Intervention	Gut Microbiota Modulation	Disease	Year of Study	Ref.
BBR, *Bifidobacterium*	↑ *R. gnavus* and *R. torques*, ↓ *K. pneumoniae*	Hyperglycemia	2015–2018	[23]
BBR, probiotic	↑ *B. breve* and *Eggerthella lenta*, ↓ *Bifidobacterium*	T2D	2016–2017	[24]
BBR, probiotic	↓ *Ruminococcus bromii*; bile acid metabolism alterations	T2D	2016–2017	[25]
BBR	↓ *Firmicutes,* ↓ *Coliforms*, ↑ *Bacteroides*	Schizophrenia, bipolar, psychosis	2018–2020	[26]
Berberine hydrochloride	↑ Bacterial diversity (Chao and Ace indices)	Parkinson’s	2021	[27]

BBR, berberine.

**Table 2 pharmaceuticals-18-00193-t002:** Effect of berberine on inflammatory biomarkers.

Aim of Study	Patient Conditions	Results	Ref.
To assess the effect of berberine on CRP levels.	Acute coronary syndrome, hypertension, T2D, obesity, and ischemic stroke	CRP: −0.64 mg/L (IC 95%: −0.67 to −0.61) *p* < 0.001	[73]
To assess the effects of berberine on the metabolic profiles of patients with type 2 diabetes.	T2D or prediabetes	CRP: (−2.13 mg/L, IC 95% −2.98, −1.28) IL-6: (−1.83 pg/mL, IC 95% −3.05, −0.61) TNF-α: −1.44 pg/mL, IC 95% −2.72, −0.16) HbA1c: (−0.73%; IC 95%: −0.97, −0.51). Fasting plasma glucose: (−0.86 mg/dL, IC 95% −1.10, −0.62). *p* < 0.00001)	[74]
To assess the effects of berberine on inflammatory markers in Chinese patients with metabolic syndrome.	Metabolic syndrome and related disorders	CRP: (−1.54 mg/L, IC 95% −1.86, −1.22), *p* < 0.05 TNF-α: (−1.02 pg/mL, IC 95% −1.27, −0.77), *p* < 0.05 IL-6: (−1.17 pg/mL, IC 95% −1.53, −0.81), *p* < 0.05 IL-1β: −0.81 pg/mL; IC 95% −1.80, 0.17), *p* = 0.11	[75]
To assess the clinical efficacy of berberine in the treatment of acute ischemic stroke (AIS).	Acute ischemic stroke	hs-CRP: (−1.33 mg/L; IC 95% −2.15, −0.51), *p* = 0.001 MIF: (−2.00 ng/mL; IC 95% −2.34, −1.67), *p* < 0.00001 IL-6: (−2.94 pg/mL; IC 95% −5.47, −0.41), *p* = 0.02 Complement C3: (−110.78 mg/dL; IC 95% −221.18, −0.37). HIF-1α: (−256.03 pg/mL; IC 95% −321.21, −190.86), *p* < 0.00001 Caspase-3: (−2.55 pg/mL; IC 95% −4.21, −0.88), *p* = 0.003	[76]
To assess the effect of berberine and barberry supplementation on inflammatory biomarkers in adults and the dose-response relationship.	Various diseases	IL-6: (−1.18 pg/mL; IC 95% −1.8, −0.55), *P* < 0.0001 TNF-α: (−3.72 pg/mL; IC 95% −5.55, −1.9), *p* < 0.0001 CRP: (−1.33 mg/L; IC 95% −1.9, −0.75), *p* < 0.0001	[77]

CRP: C-reactive protein; TNF-α: tumor necrosis factor-alpha; IL-6: interleukin 6; IL-1β: interleukin 1 beta; HbA1c: glycosylated hemoglobin; hs-CRP: high-sensitivity C-reactive protein; MIF: macrophage inhibitory factor of migration; HIF-1α: hypoxia-inducible factor 1-alpha; T2D: type 2 diabetes.

## Data Availability

No new data were created or analyzed in this study. Data sharing is not applicable to this article.

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
