# Peer review of "Perspectives on Berberine and the Regulation of Gut Microbiota: As an Anti-Inflammatory Agent"

_pharmaceuticals, 2025, doi:10.3390/ph18020193_

Round 1

Reviewer 1 Report

Comments and Suggestions for Authors

Dear Authors.

Your extensive research paper on the medicinal properties of berberine is undoubtedly very well written and has a sufficient number of scientific sources that you rely on in your research.

Berberine has been shown to have anti-inflammatory, antimicrobial, anti diabetic, and protective effects on the intestinal barrier. Resent studies have shown that Berberine modulates the intestinal microbiota y increasing the frequency of beneficial bacteria, thereby improving metabolic health and reducing inflammation. Furthermore, when combined with probiotics, it appears to enhance their useful effects, suggesting that it could be used as a prebiotic agent in the treatment of various clinical conditions, in IBS, IBD, UC reduces inflammation and improvement of intestinal microbiota, leading to reducing of symptoms.

Without a doubt, your work is worthy of being published in such a respected  Journal. However, I would like to make some comments and clarifications to the text:

1. Not a single word is said about natural sources of berberine in nature. But many very effective folk tinctures and decoctions are based on various extracts from plant matrices. For example, the berberine-containing extract from Phellodendron amurense, used in South-East Asia, etc.

2. In this regard, I think it is necessary to make a short chapter specifically on folk remedies based on berberine-containing products, and a brief description of their confirmed effect.

3. Due to the fact that berberine has a confirmed effect on the body mass index and glycated hemoglobin, it is possible to also describe these studies in more detail and highlight them in a separate additional chapter. Since obesity is a problem for all of humanity, and the fight against it will not always be successful. This point will also be extremely interesting in further pharmaceutical developments.

Author Response

Comments 1: [Not a single word is said about natural sources of berberine in nature. But many very effective folk tinctures and decoctions are based on various extracts from plant matrices. For example, the berberine-containing extract from Phellodendron amurense, used in South-East Asia, etc.]
Response 1: Thank you for your suggestion. In the revised version of the manuscript, we have included a section on the natural occurrence of berberine in various plants, including Phellodendron amurense, 1.1 Berberine pages 1-2, lines 29-90 . We have also emphasized the significance of these plant extracts within the context of folk medicine. Additionally, we included a new section in the introduction 1.2 Traditional use of plant species containing berberine on page 3, lines 92-113.
Comments 2: [In this regard, I think it is necessary to make a short chapter specifically on folk remedies based on berberine-containing products, and a brief description of their confirmed effect]
2
Response 2: We appreciate your recommendation to enrich the manuscript. Considering the current length of the text and the work's focus, we decided to incorporate a section entitled 1.2 Traditional use of plant species containing Berberine. This section is on page 3 lines 92-113.
Comments 3: [Due to the fact that berberine has a confirmed effect on the body mass index and glycated hemoglobin, it is possible to also describe these studies in more detail and highlight them in a separate additional chapter. Since obesity is a problem for all of humanity, and the fight against it will not always be successful. This point will also be extremely interesting in further pharmaceutical developments]
Response 3: We appreciate your insightful comments and suggestions. We agree that the impact of berberine on body mass index and glycated hemoglobin is highly significant, especially given the global challenge of obesity. The two new sections are 4.0 Effects of Berberine on Obesity, on pages 12-13 lines 487-541, and 4.1 Effect of Berberine on Diabetes, on pages 13-14, lines 543-595.
4. Response to Comments on the Quality of English Language
Point 1:
Response 1: (in red)
5. Additional clarifications
[Here, mention any other clarifications you would like to provide to the journal editor/reviewer.]

Reviewer 2 Report

Comments and Suggestions for Authors

The review article on the pharmaceutical potential of Berberine submitted for review contains a large dose of literature information. The text is arranged logically, but being a review of literature information, it omits the first chapter, which should be and contain information about Berberine: Structural formula, chemical name, physicochemical properties, sources of extraction (barberry varieties and parts of the plant used, method of extraction, purification), annual production, price per kg or ton.

            Berberine has been used in traditional medicine for hundreds of years, so it is very surprising that the literature review is limited only to works from the 21st century, as if there were no earlier ones. I understand that modern computerized sources are readily available, but earlier ones do exist and probably contain valuable information as well. A well-prepared literature review should also address them.

            In their conclusions, the Authors should also refer to potentially dangerous metabolites - e.g. a structure devoid of saturated binding at the nitrogen atom. Can such a metabolite be formed in the oxidation process, how will it affect the photoinduced processes in the skin? It should fluoresce strongly and potentially sensitize to light.

            The Authors write about berberine nanoparticles (e.g. in line 295), but do not give their size, which has a large impact on bioavailability. This should be supplemented as much as possible.

            After supplementing the text with the above-mentioned deficiencies, the article, in my opinion, can be published.

Author Response

Comments 1: [The review article on the pharmaceutical potential of Berberine submitted for review contains a large dose of literature information. The text is arranged logically, but being a review of literature information, it omits the first chapter, which should be and contain information about Berberine: Structural formula, chemical name, physicochemical properties, sources of extraction (barberry varieties and parts of the plant used, method of extraction, purification), annual production, price per kg or ton.]
Response 1: We appreciate your suggestion. We have included information that addresses your comment in the revised version of the manuscript. The additional information can be found in the
2
Introduction section, 1.1 Berberine, pages 1,2, lines 29-90. The chemical structure of Berberine is in Figure 1.
Comments 2: [In this regard, I think it is necessary to make a short chapter specifically on folk remedies based on berberine-containing products, and a brief description of their confirmed effect]
Response 2: We appreciate your recommendation to enhance the manuscript. Considering the current length of the text and the project's focus, we have decided to include a section titled 1.2 Traditional use of plant species containing Berberine. This section is on page 3 lines 92-113.
Comments 3: [Berberine has been used in traditional medicine for hundreds of years, so it is very surprising that the literature review is limited only to works from the 21st century, as if there were no earlier ones. I understand that modern computerized sources are readily available, but earlier ones do exist and probably contain valuable information as well. A well-prepared literature review should also address them]
Response 3: We appreciate your comment and acknowledge that information about Berberine published before the 21st century exists. However, our focus for this paper found that the most relevant information, in terms of academic rigor and scientific understanding, is predominantly from this century. The availability of information for most readers is crucial when preparing a review.
It is important to note that using recent references reflects the scientific community's current interest in specific topics. This makes the review more engaging, emphasizing state-of-the-art information and encouraging more profound research. Recent references also help authors establish trust with readers regarding the reliability of the information presented.
In fact, we have included pioneering papers related to this review's focus. Additionally, readers can find references to information published before this century in the citations provided
Comments 4: [ In their conclusions, the Authors should also refer to potentially dangerous metabolites - e.g. a structure devoid of saturated binding at the nitrogen atom. Can such a metabolite be formed in the oxidation process, how will it affect the photoinduced processes in the skin? It should fluoresce strongly and potentially sensitize to light.]
Response 4: Thank you for your comment. In the revised manuscript, we added Section 4.2, Toxicity of Berberine, on page 14, lines 597-615. This section discusses the toxicity and effects of high oral dosages of Berberine. Our conclusions on this matter are presented in Section 5, spanning page 15, lines 636-641.
Comment 5: [The Authors write about berberine nanoparticles (e.g. in line 295), but do not give their size, which has a large impact on bioavailability. This should be supplemented as much as possible.]
Response 5: Thank you for your comment. We have included the particle size (nm) and the hydrodynamic average size of berberine nanoparticles on Page 8, lines 376-379.
3
4. Response to Comments on the Quality of English Language
Point 1:
Response 1: (in red)
5. Additional clarifications
[Here, mention any other clarifications you would like to provide to the journal editor/reviewer.]

Round 2

Reviewer 2 Report

Comments and Suggestions for Authors

The submitted version of the article is much better than the original one. Information about berberine was supported by new ones, which made it necessary to rewrite many chapters.

I have no critical comments about the current version of the article and I suggest its publication.

The only objection concerns the structural formula, on which the marked locant numbers (numbers of selected atoms 51, 49, 48 and 46) coincide or are too far from the symbols of bonds and atoms. The drawing, in this respect, needs improvement.

Author Response

Comments 1: [The submitted version of the article is much better than the original one. Information about berberine was supported by new ones, which made it necessary to rewrite many chapters.

 I have no critical comments about the current version of the article and I suggest its publication.

The only objection concerns the structural formula, on which the marked locant numbers (numbers of selected atoms 51, 49, 48 and 46) coincide or are too far from the symbols of bonds and atoms. The drawing, in this respect, needs improvement..]

Response 1: Thank you for your suggestion. In the revised version of the manuscript, you will find the corrected structural formula of Figure 1 on page 2.
